# High-Frequency ac Susceptibility of Iron-Based Superconductors

**DOI:** 10.3390/ma15031079

**Published:** 2022-01-29

**Authors:** Gianluca Ghigo, Michela Fracasso, Roberto Gerbaldo, Laura Gozzelino, Francesco Laviano, Andrea Napolitano, Guang-Han Cao, Michael J. Graf, Ruslan Prozorov, Tsuyoshi Tamegai, Zhixiang Shi, Xiangzhuo Xing, Daniele Torsello

**Affiliations:** 1Department of Applied Science and Technology, Politecnico di Torino, 10129 Torino, Italy; michela.fracasso@polito.it (M.F.); roberto.gerbaldo@polito.it (R.G.); laura.gozzelino@polito.it (L.G.); francesco.laviano@polito.it (F.L.); andrea.napolitano@polito.it (A.N.); daniele.torsello@polito.it (D.T.); 2Istituto Nazionale di Fisica Nucleare, Sezione di Torino, 10125 Torino, Italy; 3Department of Physics, Zhejiang University, Hangzhou 310027, China; ghcao@zju.edu.cn; 4Department of Physics, Boston College, Chestnut Hill, MA 02467, USA; grafm@bc.edu; 5Ames Laboratory, Ames, IA 50011, USA; prozorov@iastate.edu; 6Department of Physics & Astronomy, Iowa State University, Ames, IA 50011, USA; 7Department of Applied Physics, The University of Tokyo, Hongo, Bunkyo-ku, Tokyo 113-8656, Japan; tamegai@ap.t.u-tokyo.ac.jp; 8School of Physics, Southeast University, Nanjing 211189, China; zxshi@seu.edu.cn; 9School of Physics and Physical Engineering, Qufu Normal University, Qufu 273165, China; xzxing@qfnu.edu.cn

**Keywords:** iron-based superconductors, high-frequency AC susceptibility, microwave superconductivity

## Abstract

A microwave technique suitable for investigating the AC magnetic susceptibility of small samples in the GHz frequency range is presented. The method—which is based on the use of a coplanar waveguide resonator, within the resonator perturbation approach—allows one to obtain the absolute value of the complex susceptibility, from which the penetration depth and the superfluid density can be determined. We report on the characterization of several iron-based superconducting systems, belonging to the 11, 122, 1144, and 12442 families. In particular, we show the effect of different kinds of doping for the 122 family, and the effect of proton irradiation in a 122 compound. Finally, the paradigmatic case of the magnetic superconductor EuP-122 is discussed, since it shows the emergence of both superconducting and ferromagnetic transitions, marked by clear features in both the real and imaginary parts of the AC susceptibility.

## 1. Introduction

Since 2006, when iron-based superconductors (IBSs) were discovered [1], they raised great interest both from a fundamental point of view and for application-oriented research [2,3]. Throughout the years, new IBS “families” came into play, sharing some common intriguing features [4], such as their multiband nature [5], unconventional pairing [6,7], and nontrivial pairing symmetries [8], and for these reasons they are currently the object of an intense study effort by research groups throughout the world.

Within this framework, the investigation of their high-frequency (microwave) properties has proven to be extremely useful in the characterization of such novel materials, yielding crucial information about the mechanisms of superconductivity [9,10,11]. This can be achieved by means of several methods, critically depending on the frequency range of interest. For the GHz range, the use of resonant microwave methods guarantees high sensitivity and accuracy, though they are generally limited to one or a few frequencies [12]. Among these, resonator-perturbation methods, based on the analysis of small changes induced by the sample in the electromagnetic response of a resonator to which it is coupled, have been employed successfully [13,14,15]. In particular, in these methods, the resonator is an efficient probe of the distribution of the electromagnetic fields inside the sample, thus giving access to their response that can be expressed in terms of the complex AC magnetic susceptibility, $\chi_{ac}$.

In this work, we report on a microwave technique based on the use of a coplanar waveguide resonator (CPWR) that is suitable for the study of very small IBS single crystals, and we show how the absolute values of $\chi_{ac}$ can be obtained for thin platelets (Section 2). It is shown that a further analysis of the complex AC susceptibility data allows one to obtain the London penetration depth $\lambda_{L}$ and therefore the superfluid density, thus enabling the validation of theories and models concerning the mechanisms of superconductivity in the IBS. For this goal, the high frequency of this technique offers an advantage with respect to conventional AC susceptometry, for which the determination of $\lambda_{L}$ is challenging [16] because of the long wavelength and the geometry associated with those experiments [17], which is more suitable for the study of vortex physics and pinning [18,19,20] and of nonlinearity [21,22]. In Section 3 of this paper, we present and discuss the characterization of several IBS systems, belonging to different IBS families. They are organized in a comparison among families (Section 3.1); a comparison of systems with different types of doping within the same family (Section 3.2); an analysis of the effects of ion irradiation, introducing disorder, on the same compound (Section 3.3); and finally the characterization of a ferromagnetic superconductor (Section 3.4). The latter case is paradigmatic, since it shows the emergence of both superconductivity and ferromagnetism, which can coexist in IBSs with magnetic rare-earth elements, as is clearly evinced by features in both the real and imaginary parts of AC susceptibility.

## 2. Materials and Methods

### 2.1. Materials

The high-quality IBS single crystals investigated in this work were prepared in different laboratories. In Table 1 we report a list of the studied materials, with indications of the doping level *x* (in all cases this was close to optimal doping level), the labels that identify them throughout the work, and a short indication of the preparation method.

### 2.2. The CPWR Method

A microwave coplanar waveguide resonator (CPWR) technique was used to investigate the high-frequency AC susceptibility of small crystals. The CPWR was obtained by standard photolithography from a YBa${}_{2}$Cu${}_{3}$O${}_{7-x}$ 250-nm-thick film deposited on a MgO substrate [30]. The sample under study was coupled to the CPWR by fixing it at the center of the 350-μm-wide central stripline, far from the edges, where a quite uniform RF (radio frequency) magnetic field is generated, parallel to the CPWR surface and with amplitude of $H_{ac}\approx$ 1 Oe (see the sketch in Figure 1a). The resonance frequency of the CPWR is about f=7.9 GHz, depending slightly on temperature. A Rohde & Schwarz ZVK vector network analyzer was used to continuously measure the transmission coefficient $S_{21}$ between the two ports sketched in Figure 1a (according to a standard scheme [31]), in a frequency window of ∼40 MHz around the CPWR resonance, while the system is slowly heated from the base temperature to about 20–30 K above the $T_{c}$ of the sample. Based on $S_{21}$ vs. frequency curve, the resonance frequency $f_{0}$ and the unloaded quality factor $Q_{0}$ are extracted through fitting with a modified Lorentzian function [15]. Within a resonator-perturbation approach [32], the fractional shifts of the resonance frequency and of the quality factor of the CPWR, due to the presence of the sample, are connected to its complex magnetic susceptibility (see Figure 1b,c) as follows:(1)$$2\frac{\Delta f_{0}}{f_{0}}+i\Delta \left(\frac{1}{Q_{0}}\right)\approx -\left(\Gamma_{f}\chi_{ac}^{\prime }+i\Gamma_{Q}\chi_{ac}^{\prime \prime }\right),$$
where the geometrical factors $\Gamma_{f}$ and $\Gamma_{Q}$ depend on the distribution of the RF fields around the resonator and on the sample shape and dimensions. Equation (Equation 1) was rewritten [33,34] based on the relations given in [32], with the approximation of a very small sample positioned in a region with a negligible electric field. These approximations affect in a different way the real and imaginary parts of Equation (Equation 1), necessitating the adoption of two slightly different geometrical factors, $\Gamma_{f}$ and $\Gamma_{Q}$ (usually within 20–25%) [33,34].

### 2.3. Calibration

The geometrical factors in Equation (Equation 1) are determined though a calibration procedure based on data collected at temperatures $T>T_{c}$ [34]. Here, the sample is supposed to behave like a metal, and the theory of electrodynamics in normal metals [35] gives, for a slab of thickness 2c, with $H_{ac}$ parallel to the slab surface:(2)$$\chi_{ac}^{\prime }=-1+\frac{\delta }{2c}\frac{\sin h\left(\frac{2c}{\delta }\right)+\sin\left(\frac{2c}{\delta }\right)}{\cos h\left(\frac{2c}{\delta }\right)+\cos\left(\frac{2c}{\delta }\right)}$$
(3)$$\chi_{ac}^{\prime \prime }=\frac{\delta }{2c}\frac{\sin h\left(\frac{2c}{\delta }\right)-\sin\left(\frac{2c}{\delta }\right)}{\cos h\left(\frac{2c}{\delta }\right)+\cos\left(\frac{2c}{\delta }\right)}$$
where $\delta =\sqrt{2\rho /\omega \mu_{0}}$ is the classical skin depth, and ρ is the DC resistivity. The complex susceptibility of a metallic slab is shown in Figure 2a as a function of the ratio δ/c.

Above $T_{c}$, for the small crystals under study, we assume that the temperature dependence of the $f_{0}$ and $1/Q_{0}$ shifts in the CPWR are mainly due to the temperature dependence of the skin depth, neglecting other contributions (e.g., from thermal expansion). Thus, we fit the experimental $f_{0}$ and $1/Q_{0}$ shifts above $T_{c}$ by means of Equations (Equation 1)–(Equation 3), with the constraint of keeping the same δ(T) for both the curves. A small correction (of the order of few percent) was also used to take into account the penetration of RF fields from the other faces of the crystal (which, due to its finiteness, is not exactly an infinite slab). With this aim, we added two further terms to Equation (Equation 2), both equal to the second term of $\chi_{ac}^{\prime }$ but with the substitution of *c* with *a* and *b*, respectively [34,36]. The same was carried out for Equation (Equation 3), where we added two terms similar to that shown in Equation (Equation 3) itself, but with *a* and *b* instead of *c*. This is a reasonable approximation when the two dimensions other than the thickness are much larger than δ (which is expected to be of the order of microns), as for our platelets. Thus, the geometrical factors $\Gamma_{f}$ and $\Gamma_{Q}$ can be obtained via a fit (as shown in Figure 1b), and kept valid in the whole temperature range, i.e., also below $T_{c}$.

Finally, through Equation (Equation 1), the absolute values of $\chi_{ac}^{\prime }$ and $\chi_{ac}^{\prime \prime }$ are also obtained for the superconducting state (Figure 1c). It can be noted that, in contrast with low-frequency $\chi_{ac}\left(T\right)$, both real and imaginary parts of the AC susceptibility assume non-zero values in the normal state, and that no peaks emerge in $\chi_{ac}^{\prime \prime }$. This is a consequence of the fact that the penetration depth (discussed in the next section) connects, at $T_{c}$, to the skin depth δ(T), which in this frequency range is lower than the half-thickness of the sample, thus resulting in shielding ($\chi_{ac}^{\prime }<0$) as well as dissipation ($\chi_{ac}^{\prime \prime }>0$) in the normal state.

### 2.4. Penetration Depths and Superfluid Density

Once the complex susceptibility has been determined in the superconducting state, the London penetration depth, $\lambda_{L}$ (corresponding to the skin depth for a superconductor), and the quasiparticle conductivity, $\sigma_{1}$, can be found. In fact, in a superconductor, the complex susceptibility $\chi =\chi_{m}+\chi_{s}$ depends on both a bulk contribution, $\chi_{m}$, and on the screening effects of supercurrents [33], generating a contribution $\chi_{s}$ that, for a slab of thickness 2c, reads:(4)$$\chi_{s}^{\prime }\approx -\left\{1-ℜ\left[\frac{\tanh(\kappa c)}{\kappa c}\right]\right\}\;;\;\chi_{s}^{\prime \prime }\approx ℑ\left[\frac{\tanh(\kappa c)}{\kappa c}\right]$$
where
(5)$$\kappa =\sqrt{1/\lambda_{L}^{2}+i\omega \mu_{0}\sigma_{1}}.$$

Thus, in the case of pure superconductors without significant magnetic ordering ($\chi_{m}\approx 0$), both $\lambda_{L}$ and $\sigma_{1}$ can be deduced by means of Equations (Equation 1), (Equation 4) and (Equation 5). On the other hand, if $\chi_{m}$ cannot be neglected, as in the case of magnetic superconductors (see below, Section 3.4), it can be determined by means of the same equations, once the contribution due to screening by supercurrents has been subtracted.

An example of skin depth (for the normal state) and penetration depth (for the superconducting state) is shown in Figure 2b. Based on the latter, the superfluid density can be calculated as $\rho_{s}\left(T\right)=\lambda_{L}{\left(0\right)}^{2}/\lambda_{L}{\left(T\right)}^{2}$.

### 2.5. Irradiation Experiments

We show in this work some results obtained on irradiated samples. Irradiations of BaCo-122 crystals with 3.5-MeV protons were performed at the INFN-Laboratori Nazionali di Legnaro, Italy, at room temperature and with the proton beam parallel to the crystal *c*-axis. Several fluences were used, up to 1.67×10${}^{17}$ cm${}^{-2}$. The 320-MeV Au ion irradiation of SeTe-11 was performed using tandem accelerator at the Japan Atomic Energy Agency, Japan, at a fluence of 10${}^{11}$ cm${}^{-2}$, with the beam parallel to the crystal *c*-axis.

## 3. Results and Discussion

In this Section, an overview of results obtained using the CPWR method described in Section 2.2 is given. The results are organized in a comparison among IBS families, among types of doping, among pristine and irradiated samples, and finally between pure superconductors and magnetic superconductors.

### 3.1. Comparison among Different IBS Families

In Figure 3 we report a comparison among the complex susceptibility of samples representing different IBS families (frame (a)), including 11, 122, 1144, and 12442 systems. Frames (b) and (c) show, for the same samples, the London penetration depth and the superfluid density.

This overview clarifies the reason that high-frequency $\chi_{ac}$ measurements and the CPWR technique in particular are valuable tools for the investigation of novel materials. In fact, the knowledge of the absolute value of $\lambda_{L}$ and/or of $\rho_{s}\left(T\right)$ for truly single crystals (which could be very small) enables the validation of theories or models concerning fundamental aspects of superconductivity in these compounds, including the coupling mechanism [37], the number and value of superconducting gaps [38], the symmetry of the order parameter [11], and $\lambda_{L}$ anisotropy [39]. In fact, though these systems share common building blocks, they show specific behaviors. As for the common features, it can be noted that all these materials are essentially based on the presence of Fe${}_{2}$X${}_{2}$ planes—where X is either a pnictogen (e.g., As) or a chalcogen (e.g., Se)—which are the layers supporting superconductivity, separated by insulating or blocking layers. The latter define the different IBS “families”, ranging from non-existing (as in the 11 family, SeTe-11 in our case), to monoatomic layers (as in the 122 family, e.g., doped BaFe${}_{2}$As${}_{2}$), to alternate monoatomic (as in the stochiometric 1144 family, CaK-1144 in our case), and to more complex structures (as in the 12442 family, Rb-12442 in our case). The importance of the blocking layers on the properties of IBS is particularly evident when looking at the differences between the BaK-122, CaK-1144, and Rb-12442 in Figure 3: these samples are close to the same carrier concentration given by K and Rb atoms [40,41]. These species are either mixed with Ba (in the 122) or ordered in layers (in the 1144 and 12442) but play the same role, yielding the same doped Fe${}_{2}$As${}_{2}$ layers. The main difference among them is therefore directly given by the spacer layer and results in quite different critical temperature, gap values and even order parameters [38,42,43].

The specifics of each system have been investigated [38,42,43], showing that 122 and 1144 families can be understood within a $s_{\pm }$-multigap model, whereas nodal (d−wave type) multigap superconductivity has been found for Rb-12442 [43]. On the other hand, SeTe-11 shows an unusually large $\lambda_{L}\left(0\right)$ value, possibly indicating a large level of disorder even in high-quality single crystals as observed by several techniques [44,45,46]. Efforts to frame the $\lambda_{L}\left(T\right)$ data of SeTe-11 within an overall theoretical picture are currently ongoing.

### 3.2. Effects of Different Kinds of Doping in Ba-122

Figure 4a shows the complex susceptibility of BaFe${}_{2}$As${}_{2}$-based systems, where superconductivity was induced by means of hole doping (BaK-122), electron doping (BaCo-122), and chemical pressure (BaP-122). Frames (b) and (c) show the London penetration depth and the superfluid density, respectively, for the same samples. It can be noticed that the $\rho_{s}\left(T\right)$ curve of the K-doped system is qualitatively very different from the curves for the other doping types, which are quite similar to one another. We showed that all these compounds can be well described within the same $s_{\pm }$-multigap model [38], and speculated that the main difference between P and Co doping with respect to K doping depends on whether the chemical substitution is performed on the FeAs planes, which are the main planes responsible for the superconducting properties (such as for P and Co doping), or outside them (such as for K doping). This affects carrier scattering and the pnictogen position (height from the FeAs plane), which are different in the two cases [47].

### 3.3. Effects of Proton Irradiation

Figure 5 shows the effects of 3.5-MeV proton irradiation on BaCo-122 crystals. This process produces defects at the nanoscale (mainly vacancies, interstitials, and small clusters) that act as efficient carrier-scattering centers [48,49].

The disorder induced by irradiation causes a decrease in $T_{c}$ and an increase in the normal state resistivity, which is clearly visible in the $\chi_{ac}\left(T\right)$ data in (a), and a dramatic increase in the penetration depth (panel (b)). Such a huge modification of $\lambda_{L}\left(0\right)$ (of the order of 2300% for the highest proton dose) despite the relatively low degradation of $T_{c}$ (about −30%) is typical of systems with short coherence lengths, where the order parameter is strongly suppressed in the vicinity of defects but mostly unaffected elsewhere [50,51]. Moreover, a similar behavior was successfully reproduced within a multiband Eliashberg approach with disorder for another similar 122 system (Ba(Fe${}_{1-x}$Rh${}_{x}$)${}_{2}$As${}_{2}$) that was irradiated with 3.5-MeV protons [11,37]. Furthermore, the superfluid density $\rho_{s}\left(T\right)$ shown in (c) is similar to the case measured and calculated in [11,52], with a relevant change in its temperature dependence after the first irradiation, and with minor modifications for higher doses. This change is attributed to the transition from a clean to a dirty system and to the relative modification of the superconducting gap values.

### 3.4. Flux-Pinning Effects

AC susceptibility can be used to study vortex dynamics and to test the flux-pinning capability of IBSs. In fact, a source of dissipation affecting the value of $\chi_{ac}^{\prime \prime }$ is the motion of vortices, which can be hampered by pinning due to defects. Figure 6 shows for SeTe-11 the fractional increment of $\chi_{ac}^{\prime \prime }$ as a function of the applied DC magnetic field, i.e., for increasing levels of vortex density. For the pristine crystal, the field dependence of this dissipation term is linear, in agreement with other literature findings for this material in this field range (e.g., for vortex-motion-induced surface resistance vs. DC magnetic field at 6 K, 16.4 GHz, and 26.6 GHz [53]). The question arises whether dissipation induced by this mechanism can be controlled and reduced by means of suitable artificial defects. Heavy-ion irradiation is a powerful tool to create columnar or at least linearly correlated defects, which proved to be very efficient, pinning centers in DC and at low frequencies. There is a lack of information about their pinning efficiency in IBSs at high frequencies, such as those investigated in this work. Thus, we irradiated SeTe-11 crystals with 320-MeV Au ions at the fluence of $10^{11}$ cm${}^{-2}$. Indeed, the results reported in Figure 6 show that improved pinning in the irradiated crystal strongly reduces the vortex-induced increment of $\chi_{ac}^{\prime \prime }$ with the DC field.

### 3.5. The Ferromagnetic Superconductor EuP-122

High-frequency susceptibility is particularly relevant for the study of systems showing the coexistence of superconductivity and ferromagnetism. In IBSs, it was shown that magnetic moments in the FeAs layers are ordered in an antiferromagnetic (AFM) configuration, and that AFM spin fluctuations induce the $s_{\pm }$ pairing at the origin of superconductivity in these materials [8]. IBS compounds containing magnetic rare-earth elements, such as the EuFe${}_{2}$As${}_{2}$-based systems, in addition possess local magnetic moments that can result in further magnetic ordering. In fact, when superconductivity is induced by chemical pressure through the isovalent substitution of P in the site of As (EuFe${}_{2}$(As${}_{1-x}$P${}_{x}$)${}_{2}$), the magnetic moments of the Eu${}^{2+}$ ions, initially in AFM order, cant out of the ab plane, yielding a net ferromagnetic component along the *c* direction [54] that, coexisting with superconductivity [55], induces a rich spontaneous vortex physics [56,57,58]. The onset of this ferromagnetic order at the temperature $T_{m}$, and typical transitions due to its dynamics, at temperatures $T_{L}$ and $T_{H}$, can be recognized clearly in the $\chi_{ac}$ features shown in Figure 7a [33,59].

### 3.6. Comparison with Other Techniques

After showing results obtained by the CPWR technique with several IBS compounds, in this subsection we propose a comparison of selected CPWR measurements with an analysis performed by means of other more standard techniques, with the twofold aim of validate the CPWR method and of highlighting its advantages.

Figure 7d shows a comparison between $\chi_{ac}^{\prime \prime }$ and the DC electrical resistance *R*, measured as a function of temperature on BaP-122 single crystals. The critical temperature of the superconducting transition, as evaluated by the two techniques, is clearly the same. Moreover, the slopes of the curves for $T>T_{c}$ are in agreement with increasing DC resistivity ρ and skin depth $\delta ∼\sqrt{\rho }$ (at high frequencies, $\chi_{ac}$ is still informative about screening and dissipation since δ is lower than the half-thickness of the crystals, as discussed in Section 2.3). For $T<T_{c}$, although the DC resistance drops to zero as the first superconducting percolation path sets up, $\chi_{ac}^{\prime \prime }$ gives information about quasiparticle-induced dissipation. In fact, the quasiparticle conductivity can be calculated from $\chi_{ac}$ in the superconducting state [34] (not shown). A further practical advantage of the CPWR technique against DC resistance is that the former is contactless.

Figure 7b,c show the complex AC susceptibility measured at the frequency of 995 Hz on EuP-122 with a standard technique (i.e., by means of a Quantum Design MPMS3 commercial susceptometer), that should be compared to the CPWR measurement reported in Figure 7a. The comparison shows how both techniques are needed to achieve a comprehensive understanding of the underlying physics, and the advantages of the CPWR method. The standard susceptibility revealed a complex scenario for this material, with several features indicating magnetic structures, below the superconducting $T_{c}$ (note that $T_{c}$ is slightly different in Figure 7a,b—black arrows—because the samples are a single crystal in (a) and non-oriented powders in (b), with slightly different P doping). There is a correspondence among such features in CPWR and standard measurements (colored arrows), although the structures are much clearer in the high-frequency analysis. However, it is from the whole set of measurements collected at very different frequencies, spanning over several decades, that a comprehensive picture about dynamical mechanisms at work in this system can be drawn. This kind of systematic analysis, over a large set of experimental conditions, is currently in progress on EuP-122 powder samples.

## 4. Conclusions

In summary, we presented a novel approach to measure the AC magnetic susceptibility of small samples in the GHz frequency range, exploiting a CPWR technique. Systematic results were reported, to show the potential of the technique in characterizing several aspects of the physics of iron-based unconventional superconductors. A comparative analysis of an ample set of IBS systems belonging to four different families and covering all possible chemical doping methods was reported. Furthermore, we discussed the effect on the complex $\chi_{ac}$ of increasing disorder introduced via proton irradiation, and the effects of pinning by correlated defects created by heavy-ion irradiation on vortex-motion induced dissipation, at microwave frequencies. Finally, the case of a ferromagnetic superconductor (EuP-122) was analyzed as a paradigmatic example, further validating the CPWR method through the comparison with a standard low-frequency technique, and highlighting its advantages in terms of the clarity of the signal features, enabling the investigation of interesting dynamics and potential new physics in unconventional systems with peculiar superconducting and/or magnetic states.

## Figures and Tables

**Figure 1 materials-15-01079-f001:**
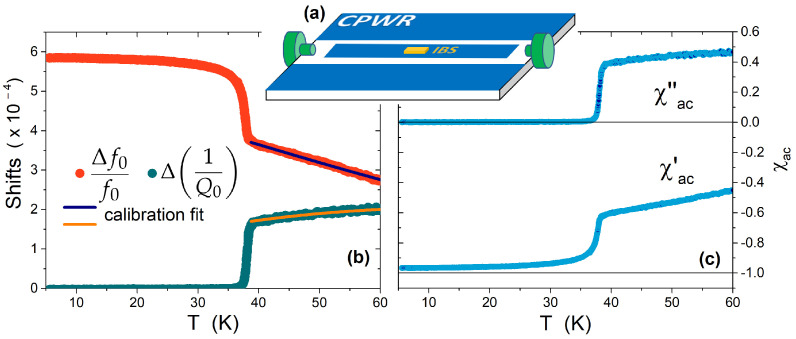
Description of the measurement method and data analysis procedure: (**a**) sketch—not to scale—of the coplanar waveguide resonator (CPWR, blue) with the IBS crystal under study (orange) coupled to it, and the launchers (green), capacitively coupling the CPWR to the measurement setup. (**b**) Shifts of the resonance frequency and of the quality factor, fitted—above $T_{c}$—by means of the procedure described in Section 2.3. (**c**) Complex AC susceptibility as a function of temperature, for a BaK-122 crystal, 11.8-μm thick.

**Figure 2 materials-15-01079-f002:**
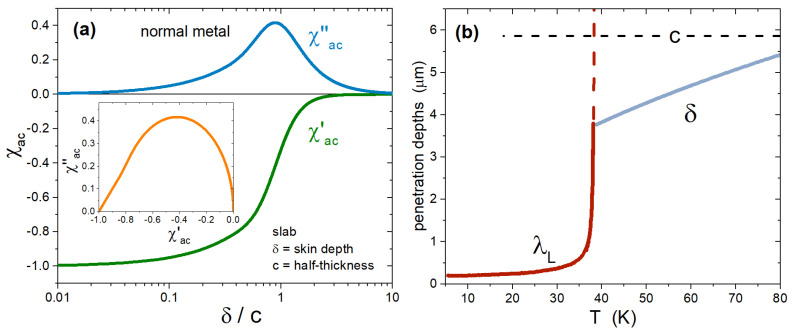
(**a**) Complex susceptibility of a metallic slab, as a function of the ratio between the skin depth δ and the half-thickness *c* of the slab, based on Equations (Equation 2) and (Equation 3). The inset shows the same data in the combination $\chi_{ac}^{\prime \prime }$ vs. $\chi_{ac}^{\prime }$. (**b**) London penetration depth $\lambda_{L}$ and skin depth δ, as a function of temperature, for a BaK-122 crystal.

**Figure 3 materials-15-01079-f003:**
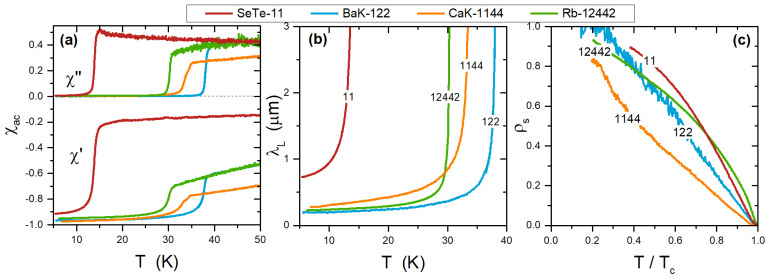
Comparison among different IBS families. Panel (**a**) shows the complex AC susceptibility of four different IBS families, 11, 122, 1144, and 12442. The penetration depth resulting from the analysis of $\chi_{ac}$ is shown in (**b**). The corresponding superfluid density is shown in (**c**).

**Figure 4 materials-15-01079-f004:**
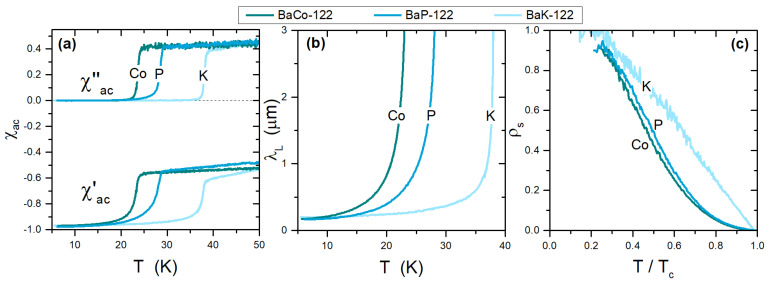
Comparison among different types of doping of the BaFe${}_{2}$As${}_{2}$ compound, with Co, P, and K. The complex AC susceptibility is shown in (**a**), the penetration depth resulting from the analysis of $\chi_{ac}$ in (**b**), and the corresponding superfluid density in (**c**).

**Figure 5 materials-15-01079-f005:**
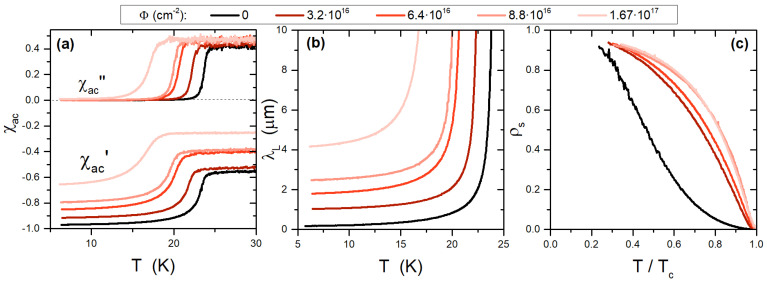
Effects of 3.5-MeV proton irradiation on BaCo-122 crystals, for different irradiation fluences, Φ. (Panel **a**) shows the complex AC susceptibility of pristine and irradiated samples, (**b**) the penetration depth, and (**c**) the corresponding superfluid density.

**Figure 6 materials-15-01079-f006:**
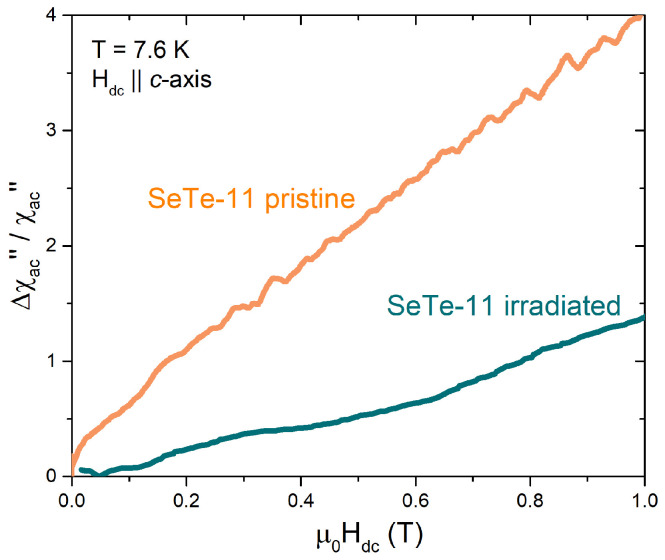
Fractional shift of the imaginary part of the AC susceptibility, $\Delta \chi_{ac}^{\prime \prime }/\chi_{ac}^{\prime \prime }=\left[\chi_{ac}^{\prime \prime }\left(H\right)-\chi_{ac}^{\prime \prime }\left(0\right)\right]/\chi_{ac}^{\prime \prime }\left(0\right)$, as a function of the applied DC magnetic field, for a pristine SeTe-11 crystal and for a SeTe-11 crystal after irradiation with 320-MeV Au ions at the fluence of $10^{11}$ cm${}^{-2}$. Data were measured at a fixed temperature of T= 7.6 K, with the field applied parallel to the *c*-axis and to the columnar defects induced by irradiation.

**Figure 7 materials-15-01079-f007:**
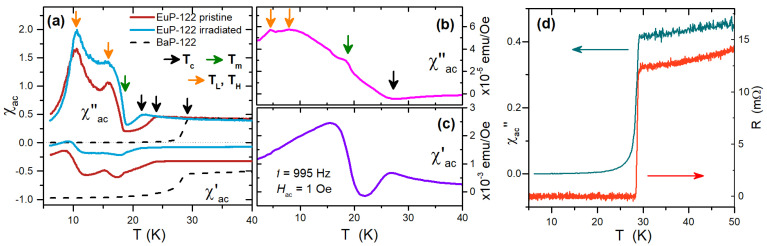
(**a**) Complex AC susceptibility of pristine and irradiated EuP-122 and BaP-122 samples with relevant temperatures marked by colored arrows. (**b**,**c**) Imaginary and real parts of the complex AC susceptibility measured at the frequency of 995 Hz by means of a standard technique, for EuP-122 powders. The arrows have the same color code as in (**a**). (**d**) comparison between the imaginary part of $\chi_{ac}$ (left scale) and the DC resistance (right scale) of BaP-122 single crystals.

**Table 1 materials-15-01079-t001:** List of the IBS materials investigated in this work, with indications of the doping level *x*, of the label that identifies each of them in this paper, and of the preparation methods, with references to the papers where further preparation details can be found.

Material	*x*	Label	Preparation Method	Ref.
FeSe${}_{1-x}$Te${}_{x}$	0.61	SeTe-11	Slow cooling—low T annealing	[23]
Ba${}_{1-x}$K${}_{x}$Fe${}_{2}$As${}_{2}$	0.42	BaK-122	FeAs self-flux	[24]
Ba(Fe${}_{1-x}$Co${}_{x}$)${}_{2}$As${}_{2}$	0.075	BaCo-122	FeAs/CoAs self-flux	[25]
BaFe${}_{2}$(As${}_{1-x}$P${}_{x}$)${}_{2}$	0.33	BaP-122	Ba${}_{2}$As${}_{3}$/Ba${}_{2}$P${}_{3}$ flux	[26]
CaKFe${}_{4}$As${}_{4}$	−	CaK-1144	High-T solution growth	[27]
RbCa${}_{2}$Fe${}_{4}$As${}_{4}$F${}_{2}$	−	Rb-12442	RbAs self-flux	[28]
EuFe${}_{2}$(As${}_{1-x}$P${}_{x}$)${}_{2}$	0.20	EuP-122	Self-flux	[29]

## Data Availability

The data presented in this article will be shared on reasonable request from the corresponding author.

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
