# Peer review of "High-Frequency ac Susceptibility of Iron-Based Superconductors"

_materials, 2022, doi:10.3390/ma15031079_

Round 1

Reviewer 1 Report

      The manuscript entitled "High-frequency ac susceptibility of iron-based superconductors" is an article which presents a novel microwave-based measurement technique for characterizing iron-based superconductors (IBS). The method employs a coplanar waveguide resonator (CPWR) connected to a vector network analyzer (VNA), which measures shifts in resonant frequency and quality factor when a very small IBS single crystal is mounted on the waveguide. The real and imaginary parts of the AC susceptibility are computed indirectly based on the measured fractional shift values. This method offers an advantage over conventional susceptometry methods as it allows the measurement of the London penetration depth, skin depth and superfluid density, giving insight regarding superconductivity mechanisms in IBS and also aiding the validation of new theories and models. The authors investigated 7 single crystal IBS superconductor samples with various compositions, showing a comparison between different IBS families, the effects of doping in a single IBS system and the effect of proton irradiation. The paradigmatic case of the ferromagnetic superconductor EuFe2(As1−xPx)2 (x = 0.2) is also presented. 

      The work is divided into 4 sections. The first section starts with an introduction regarding iron-based superconductors, resonant microwave methods and the main advantages of the CPWR technique. Section 2 indicates the samples which were investigated in this work along with their compositions. Also, a description of the CPWR method along with the calibration procedure is described in this section. Section 3 presents the main results. A comparison between different IBS families is presented in section 3.1, while the effects of doping in the same IBS family are shown in section 3.2. The effects of proton irradiation are discussed in figure 3.3 and the characterization of the  EuFe2(As1−xPx)2 (x = 0.2) superconductor is presented in section 3.4. The paper ends with conclusions followed by the bibliography. 

       The topic chosen by the authors is novel and of great interest, and the presented results and references are very recent. The order of the sections and the discussions is logical and the article is very well illustrated, which supports the data presented in the text. However, the paper has several shortcomings which are described in detail below. 

      While the method and the data presented in the manuscript are very interesting, except for the FeSe1−xTex system the presented results were previously discussed and published by the authors using the same CPWR technique which is described in this work. Therefore, this paper feels more like a review on the CPWR method and IBS single crystals than a research article. I recommend changing the manuscript type from Article to Review. 

      In section 2.1, a short description or summary regarding the synthesis of each sample should be made, perhaps in tabular form. This would paint a clearer picture to the reader, as going through the references to find the synthesis method for a particular sample may be tedious. The names of the laboratories in which the samples were prepared are irrelevant as long as the references are given. 

      In section 2.2, the CPWR method should be described in more detail, for example, how were the frequency and quality factors measured, what was used as a signal source, what was used as a frequency/Q-meter (VNA, spectrum analyzer etc). In previously published works, the authors used a VNA for the measurements. I assume it was the same in this work, however, it should be specified. An electrical schematic of the measurement setup could be drawn along with the resonator. A reference should be given for relation (1). Also, the authors mention the need to use correction factors for the real and imaginary parts of equation (1) due to the approximations which were made. The correction factors should be detailed in the text (if needed , for each sample) along with a short explanation. 

      In section 2.3, the authors mention a small correction factor to take into account the penetration of RF fields from the other faces of the crystal. These factors should be detailed in the text (if needed , for each sample) along with a short explanation. Also, the authors extend relation (2) with similar contributions from the other two dimensions of the crystal. These relations should be given in the text. 

      In section 3, it would be beneficial, if possible and if the information is available, to compare the results obtained with the CPWR method with results on similar systems obtained with different methods, for example resistivity or magnetometry measurements. 

      The manuscript also has some minor issues which are outlined below:

  • section 1, line 41 "experiment" should be changed to "experiments";
  • section 2.1, line 69, the acronym RF should be defined as "radio frequency" the first time it is used;
  • section 2.1, table 1, according to reference [23], for FeSe1−xTex, x should be 0.61;
  • section 3.3, figure 5 - the parameter Φ should be defined. 

Reviewer 2 Report

Comments to Authors

In this manuscript, the authors report the response of ac susceptibility of iron-based superconductors (IBSs) in high-frequency range. The authors have used a coplanar waveguide resonator method to investigate the iron-based superconducting systems belonging to 11, 122, 1144, and 12442 families. The authors also studied the effect of disorders (produced by proton irradiation) on ac susceptibility of IBSs.

The manuscript is well organized and written. However, most of the data was collected from other papers. I am a bit worried about the novelty of the work for Materials. The authors must highlight the significance of their work. They should also showcase their progress by tabulating their results along with past advancements made in the development of efficient IBSs. They also need to add some details about coplanar waveguide resonator method which they used for this study. Moreover, their conclusion section is very similar to that of abstract; it does not include the reals conclusions of this work. The progress of this work will decide about its publication status.

Reviewer 3 Report

The authors perform analysis of the microwave technique that is suitable for the study of AC magnetic sensitivity. They  use a method based on the use of coplanar waveguide resonator, within the perturbation approach the resonator allows to achieve the absolute value of complex sensitivity, which determines the depth of penetration and can determine the superfluid density. The authors presented  an approach for measuring AC magnetic sensitivity of small samples in  the GHz frequency band using the CPWR technique. Also , superconducting ferromagnetics (EuP-122) was analyzed as a paradigmatic example of the potential of technique in the study of unconventional systems with special states that give rise to interesting dynamics.

I suggest the authors to describe the obtainer results in more detail.

Round 2

Reviewer 1 Report

 The manuscript entitled "High-frequency ac susceptibility of iron-based superconductors" is an article which presents a novel microwave-based measurement technique for characterizing iron-based superconductors (IBS). The method employs a coplanar waveguide resonator (CPWR) connected to a vector network analyzer (VNA), which measures shifts in resonant frequency and quality factor when a very small IBS single crystal is mounted on the waveguide. The real and imaginary parts of the AC susceptibility are computed indirectly based on the measured fractional shift values. This method offers an advantage over conventional susceptometry methods as it allows the measurement of the London penetration depth, skin depth and superfluid density, giving insight regarding superconductivity mechanisms in IBS and also aiding the validation of new theories and models. The authors investigated 7 single crystal IBS superconductor samples with various compositions, showing a comparison between different IBS families, the effects of doping in a single IBS system and the effects of proton irradiation. The paradigmatic case of the ferromagnetic superconductor EuFe2(As1−xPx)2 (x = 0.2) is also presented. A comparison with other standard techniques (resistivity and low-frequency AC susceptibility measurements) is made in order to further validate the CPWR approach.  

      The work is divided into 4 sections. The first section starts with an introduction regarding iron-based superconductors, resonant microwave methods and the main advantages of the CPWR technique. Section 2 indicates the samples which were investigated in this work along with their compositions and a short description regarding synthesis. Also, a description of the CPWR method, the calibration procedure, and the details regarding the irradiation experiments are described in this section. Section 3 presents the main results. A comparison between different IBS families is presented in section 3.1, while the effects of doping in the same IBS family are shown in section 3.2. The effects of proton irradiation are presented in section 3.3. Section 3.4 discusses flux pinning effects in irradiated FeSe1−xTex (x = 0.61) crystals. The characterization of the  EuFe2(As1−xPx)2 (x = 0.2) superconductor is presented in section 3.5, while a comparison of the CPWR method with other standard techniques is shown in section 3.6. The paper ends with conclusions followed by the bibliography. 

       The topic chosen by the authors is novel and of great interest, and the presented results and references are very recent. The order of the sections and the discussions is logical and the article is very well illustrated, which supports the data presented in the text. The manuscript has been greatly improved and the authors have successfully addressed all of the issues raised by the reviewer.